# Study on Preparation of Superhydrophobic Copper Surface by Milling and Its Protective Performance

**DOI:** 10.3390/ma15051939

**Published:** 2022-03-05

**Authors:** Chenxi Jia, Jiyuan Zhu, Langping Zhang

**Affiliations:** College of Mechanical and Control Engineering, Guilin University of Technology, Guilin 541004, China; jcx18353112972@163.com (C.J.); zhanglangping123@163.com (L.Z.)

**Keywords:** copper, superhydrophobic, milling, corrosion protection

## Abstract

Using milling method, a 0.1 mm flat-bottom sharp knife was used to mill the surface of Cu substrate in a CNC engraving machine to construct the microstructure of rectangular bumps, and rectangular bumps with different sizes and different distances were prepared by changing the distance between cutter tips. After deburring and stearic acid modification, a superhydrophobic Cu surface with excellent mechanical durability and stability was successfully prepared. Through friction and wear experiments, the contact angle of the superhydrophobic Cu surface decreased slightly while retaining excellent corrosion resistance.

## 1. Introduction

Cu has excellent machinability, chemical stability, and thermal conductivity, and it is widely used in various fields such as military, industry, and daily life [1]. Although Cu and its alloys are very stable in dry environments, Cu is easily corroded in humid environments, which limits its practical application [2]. Therefore, how to improve the corrosion resistance of Cu in humid environments has become a heated topic in material research [3]. Because of its great potential and application in the field of corrosion protection, superhydrophobic film has attracted extensive attention in recent years [4,5]. Relevant research shows that superhydrophobic film can greatly improve the water contact angle of the surface, hindering the contact between corrosive medium and metal surface, and providing corrosion protection for the substrate without affecting the mechanical properties of metal [6,7,8]. So far, the methods of preparing superhydrophobic surface on Cu include etching [9,10], sol–gel [11,12], and chemical vapor deposition [13,14], but the practical application of superhydrophobic surface is often limited because of poor mechanical and chemical stability [15].

Superhydrophobic surfaces are usually obtained by combining surface roughening with low surface energy modification materials [16,17]. So how to get roughened surfaces with high mechanical strength has become an urgent problem to be solved. Forooshani et al. [18] took advantage of the dislocation defects in crystalline metals which are more likely to be eroded, and prepared the superhydrophobic Cu surface with micro-nano structure by dislocation etching, shot peening, and stearic acid treatment, and the highest contact angle reached 153°. Because of the uncontrollable chemical etching surface, shot peening is used to densify the surface after etching, making for a cumbersome process to roughen the surface. Ta et al. [19] used nanosecond laser to ablate copper/brass surface to prepare superhydrophobic surfaces. The sample reached a superhydrophobic state after about 11 days, showing a problem of laser-induced superhydrophobic surfaces which often need to be placed in the air for a long time to achieve stable superhydrophobic properties. The transformation comes from the spontaneous chemical adsorption of air pollution on ablated surfaces [20], and its stability is difficult to control due to the external environment. At the same time, owing to the thermal heating or interaction with other liquid medium, such superhydrophobic coating can be easily damaged, causing the uncontrolled loss of superhydrophobic behavior [21]. Mechanical processing is one of the most effective means for stable and large-scale production. Milling is a mechanical processing method for cutting out the required shapes and features through a milling cutter rotating at a high speed on a lathe [22]. Previously, Rahman et al. [23] studied the contact angle hysteresis and sliding behavior of water droplets on the surface of micro-groove brass obtained by milling and proved that the behavior of water droplets on the surface is closely related to periodic surface grooves obtained by micro-milling.

In previous studies, we did not consider the mechanical properties of the microstructure of Cu surface. Therefore, in order to find a preparation method of superhydrophobic Cu surfaces with mechanical stability, durability, and controllable surface morphology, we decided to start with the machining method, shaping the surface microstructure by milling, controlling the morphology of the surface microstructure by changing milling parameters, and finally combining the modification of low surface energy substances to construct a superhydrophobic surface.

In this paper, regular surface microstructure was manufactured on Cu substrate by milling, and then after modifying with stearic acid, a superhydrophobic Cu surface was prepared. The prepared superhydrophobic Cu surface has excellent mechanical durability and stability. By changing the cutter tip distance, regulation and control of the surface microstructure can be realized on the milled roughened surface, which eliminates the defect that most superhydrophobic surface microstructures are random and difficult to control. At the same time, this method of preparing superhydrophobic surface by milling and modification is also suitable for other metals, which lays a foundation for future research. The experiment does not need expensive chemical reagents and complicated instruments, and the whole preparation process is simple, efficient, and environmentally friendly. In today’s increasingly developed industrialization, the large-scale use of machine tools and the emergence of more sophisticated machine tools have contributed to the technical feasibility of milling to prepare superhydrophobic surfaces at an industrial scale.

## 2. Materials and Methods

### 2.1. Materials

H62 brass (60.5–63.5% Cu, Zn ≥ 35.7%, Fe ≤ 0.15%, P ≤ 0.01%) is a common brass variety that is widely used. It has good mechanical properties, good plasticity in a hot state, plasticity in a cold state, good machinability, and it is easy to braze and weld. At the same time, it has certain corrosion resistance, but it is prone to corrosion cracking. Mechanical properties are (thickness 0.3–10, 20 °C): tensile strength (σb/MPa): 410–630, elongation (δ 10/%): ≥ 10, and Vickers hardness (HV)105–175 (thickness ≥ 0.3). H62 brass plate for the experiment was purchased from Dongguan Minghao Metal Technology Co., Ltd. (Dongguan, Guangdong, China) and cut into 15 mm × 15 mm × 5 mm samples. Stearic acid (C_18_H_36_O_2_, AR) and hydrogen peroxide (H_2_O_2_, AR) are provided by Shanghai Yien Chemical Technology Co., Ltd. (Shanghai, China), concentrated hydrochloric acid (HCl, 36%) was supplied by Changzhou Xuhong Chemical Co., Ltd. (Changzhou, Jiangsu, China). Anhydrous ethanol was purchased from Fuyu Fine Chemical Co., Ltd. (Tianjin, China), and sodium chloride (NaCl) was supplied by Xilong Scientific Co., Ltd. (AR, Shantou, China).

### 2.2. Methods

Before milling, SiC sandpaper (300#, 600#) was used to polish Cu samples, which were then rinsed with deionized water. Attention should be paid to polishing the periphery of the Cu sheet to facilitate the fixation of the sample during milling.

The milling program was written based on CAXA Manufacturing Engineer 2013, the generated tool path program is imported into the CNC engraving and milling machine (BMDX-5040, BaoMa NC equipment Co., Ltd, Suzhou, China), and the Cu sample is processed. Figure 1a is the milling flow chart. Firstly, the surface of the Cu block is roughly machined with an end-milling 4 mm in diameter (Figure 1b). The purpose of roughing is to make the sample surface and the milling plane flush and avoid the inconsistency of surface groove depth caused by flat-bottomed sharp knife finishing. At this time, the spindle speed is 11,000 r/min, the feed rate is 400 mm/min, and the milling depth is 0.1 mm. After rough machining, the work bit is replaced with a flat-bottomed sharp knife with a tip diameter of 0.1 mm (Figure 1c). After manual tool setting, the program to finish machining the Cu block surface is loaded; the spindle speed is 12,000 r/min, the feed rate is 30 mm/min, the cutting depth is 0.01 mm each time, and the total cutting depth is 0.05 mm. The different tip distances of the milling cutter were set to 0.25 mm, 0.30 mm, and 0.35 mm respectively, and finally three kinds of samples with different distances (MS-25, MS-30, and MS-35) were obtained, whose surface microstructures are shown in Figure 1d.

The experimental parameters refer to the related research of Rahman, M.A. et al. [23] and Pratap T. et al. [24] and are adjusted according to the actual situation. Because of the precision of the machine tool, the rotation speed of the spindle should not be too fast during finishing, and a faster spindle rotation speed will cause the tip of the flat-bottomed sharp knife to shake greatly, thus increasing the dimensional error of the groove and affecting the experimental results. At the same time, the feeding rate of finishing should not be too high, because a higher feeding rate will easily cause the tip of the flat-bottomed sharp knife to break in the milling process, which will affect the normal milling process.

After taking out the processed sample, the surface was rinsed with deionized water to remove the residual milling chips on the surface. Then, in order to prevent the subsequent process against contamination, the sample was put into a beaker filled with anhydrous ethanol solution and subjected to ultrasonic cleaning many times to remove the cooling liquid flowing into the surface of the Cu sample during milling. 40 mL deionized water, 10 mL H_2_O_2_, and 10 mL HCl solution (36%) were poured into the same beaker, mixed, and stirred evenly to obtain H_2_O_2_- HCl solution. The cleaned Cu sample was then put into the mixed solution. After resting for 30 s, the sample was taken out and washed in absolute ethanol solution by ultrasonic cleaning for 10 min and then dried for later use.

To completely dissolve stearic acid, 1 g stearic acid was added into a beaker with 100 mL absolute ethyl alcohol. The dried Cu sample was put into stearic acid solution for a 5 min ultrasonic treatment, and taken out after resting for 10 min. The surface of the sample was then slowly rinsed with deionized water before drying in an oven.

### 2.3. Sample Characterization

The morphologies of the samples were characterized by field emission scanning electronic microscopy (SEM, SU5000, HITACHI, 5.0 KV, Tokyo, Japan). The chemical compositions and valence states of the samples were measured using X-ray photoelectronspectroscopy (XPS, 250Xi, Thermo scientific, Waltham, MA, USA) with the Al Kα X–raysource (hν = 1486.6 eV). The chemical compositions were also determined by Fourier Transform Infrared spectroscopy (FTIR, IRAffinity-1S, Shimadzu Corporation, 4000–450 cm^−1^, Tokyo, Japan). The static contact angle (CA) was measured with a contact angle measuring instrument (SDC-200, Xindin Precision Instrument Co., Ltd., Dongguan, China). To ensure the accuracy of the test, 6μL static water droplets were placed in five different positions of each sample for contact angle testing, and the average value of five groups of contact angle values was taken.

### 2.4. Corrosion Resistance Evaluation

To study the corrosion resistance of superhydrophobic Cu samples milled with different cutter tip distances, a CS2350H electrochemical workstation (Wuhan Corrtest Instruments Corp., Ltd., Wuhan, China) was used for measurement, using a conventional three electrode set-up with a platinum plate as the auxiliary electrode, a Ag/AgCl (Saturated KCl) electrode as the reference electrode, the corrosion resistance of the material was evaluated with 3.5 wt % NaCl aqueous solution as electrolyte. With the sample as the test electrode, the exposure area was 1 cm^2^, and the tests were done at room temperature. The corrosion potential (*E_corr_*) and corrosion current density (*I_corr_*) were extracted using the Tafel extrapolation method with the aid of the software CorrView, and calculating polarization resistance (R_p_, Ω∙cm^2^). The initial potential of potential scanning was −0.5 V, the terminal potential was 1 V, and the potential interval of data acquisition was 0.5 mv. Before testing, the sample was kept in the solution for 30 min to ensure the stability of the surface.

## 3. Results and Discussion

### 3.1. Sample Characterization

Figure 2 is the SEM image of the surface of the Cu sample processed by the CNC engraving and milling machine. There are many thread-like tool marks in the grooves of the Cu surface, which are scratches produced by flat-bottom sharp knives during milling finishing (Figure 2b). After enlargement, it can be observed that there are oblique scratches on the surface of rectangular protrusions which are tool marks produced by the end-milling cutter during the milling rough machining (Figure 2f). Obviously, it can be observed that there are many blocky and strip burrs around the bumps on the Cu surface before deburring caused by vibration or wear of the machine tool (Figure 2c).

Figure 3 is the SEM image of the Cu sample surface after deburring with H_2_O_2_- HCl solution. It can be observed that, after deburring, the oblique scratches left in rough machining and the spiral scratches left from finish machining have disappeared (Figure 3a). At the same time, most of the burrs left by milling near the bumps were removed (Figure 3d), because the H_2_O_2_- HCl solution reacted with the Cu surface, and Cu was corroded, leaving only parts of the larger burrs (Figure 3e). Some of the burrs that could not be completely removed were turned into clusters and small bumps on the surface of the bumps and distributed on the surface of the samples (Figure 3c,d).

It can be seen that there is some error between the actual shape and the ideal shape of the groove. The bumps on the processed surface present a trapezoidal boss structure. This is due to the combined action of machine tool vibration and tool wear, making the actual size of the groove larger than the theoretical size. As shown in Figure 2, the theoretical value of the groove width of each machining size should be 100 μm, and after measurement, the actual groove width of the sample in Figure 2c is about 102 μm, and about 105 μm in Figure 2e.

### 3.2. Surface Wettability Analysis

The contact angles (CA) of the Cu samples treated with different cutter tip distances are shown in Figure 4. Figure 4a,b is a sample prepared with a tip distance of 0.25 mm. The CA of the surface of the sample without stearic acid modification is 117.6° (Figure 4a), and that of the sample after stearic acid modification is 154.1° (Figure 4b). In Figure 4c,d, the cutter tip distance is 0.30 mm, the CA of the unmodified sample surface is 103.7° (Figure 4c), and the CA of the stearic acid modified sample surface is 152.4° (Figure 4d). Figure 4e,f shows that the cutter tip distance during sample preparation is 0.35 mm, in which Figure 4e shows the unmodified sample with the surface CA of 99.8°, and Figure 4f shows that the sample modified by stearic acid with the surface CA is 150.1°. The surface CA of untreated H62 copper block is 88.9°. The experimental results show that the CA of the surface of the milled Cu block is obviously larger than that of the untreated Cu block, and within a certain range, the CA of the sample surface gradually increases with the decrease in the tip distance, and the CA of the milled sample surface without stearic acid modification reaches a maximum of 117.6° and a minimum of 99.8°, which meets the conditions for a hydrophobic surface.

The contact situation between the surface of the sample without stearic acid modification (Figure 4a,c,e) and the droplet accords with the Wenzel model as the air in the groove is discharged by water, and the surface is in a highly viscous state [25]. Moreover, with the increase in the tip distance, the CA gradually decreases. Figure 4b,d,f is the surface modified by stearic acid. It can be clearly observed that there are many gaps between the sample surface and the droplets, which conforms to the Cassie model. As the air in the groove is not exhausted, the CA between the sample surface and the droplets gradually decreases with the increase in the tip distance. At this time, due to the interaction between the low surface energy modified by stearic acid and the micro-bumps on the surface, the contact area between the superhydrophobic Cu surface and the surface droplets is very small. According to formula (1) for calculating the CA of water droplets at the liquid–solid–gas three-phase composite interface proposed by Cassie et al. [26],
Cosθ_r_ = f_1_cosθ − f_2_(1)
where f_1_ is the ratio of liquid to air contact, f_2_ is the ratio of solid surface to air contact, where f_1_ + f_2_ = 1, and θ_r_ and θ represent the contact angles of liquid droplets on superhydrophobic copper surface and untreated H62 brass surface respectively. According to the experimental results, θ = 88.9° and θ_r_ = 154.1°, 152.4°, and 150.1° are substituted into Equation (1), respectively. It can be known that f_1_ (150.1°) = 0.0835, f_2_ (150.1°) = 0.9165, and the contact area between the liquid drop and the solid surface accounts for about 8%, while the remaining area of about 92% is the contact area between the liquid drop and the air; f_1_ (152.4°) = 0.0714, f_2_ (152.4°) = 0.9286, the liquid–solid contact area accounts for about 7%, and the remaining 93% is the contact area between water droplets and air; f_1_ (154.1°) = 0.0630, f_2_ (154.1°) = 0.9370, the liquid–solid contact area accounts for about 6%, the contact area between water droplets and air accounts for about 94%. The results show that the surface microstructure obtained by milling and the low surface energy modified by stearic acid play a key role in the superhydrophobicity of the sample.

### 3.3. Surface Composition Analysis

Fourier transform infrared spectrometer (FTIR) was used to analyze the chemical composition of the surface of Cu samples modified by stearic acid. Figure 5a shows the FTIR spectrum of stearic acid and Figure 5b shows the FTIR spectrum of superhydrophobic Cu sample surface. As shown in the figure, the absorption peak of stearic acid at ~1700 cm^−1^ was assigned to the stretching vibration of -COOH. After the reaction between stearic acid and the sample, the free -COOH band of stearic acid at 1703 cm^−1^ disappears, and the bending vibration of -OH at 934 cm^−1^ also greatly decreases. At the same time, the peaks at 2916 cm^−1^ and 2848 cm^−1^ correspond to the stretching vibration C-H bond and asymmetric stretching vibration C-H bond in stearic acid. Therefore, it can be concluded that the CH_3_(CH_2_)_12_COO- functional group in stearic acid reacted with the surface of Cu sample, and stearic acid was successfully modified on the surface of the Cu sample.

In order to further determine the chemical composition and characteristics of the surface of superhydrophobic Cu samples, XPS analysis was carried out on the samples. Figure 6 shows the XPS full scanning spectrum of the surface of superhydrophobic Cu samples. As shown in the figure, there are C, O, Cu, and Zn elements in the samples, and the narrow scanning spectra of C 1s, O 1s, Cu 2p, and Zn 2p are shown in Figure 7a–d.

In Figure 7a, the C 1S peak can be divided into three groups of characteristic peaks. The first peak is carbon-carbon single bond (C-C), and its characteristic binding energy is 284.8 eV. The second binding energy is related to carbon-oxygen single bond (C-O) at ~286 eV, and the third peak is located at ~289 eV due to carboxyl (O-C=O). O 1s peak (Figure 7b) consists of two groups of characteristic peaks. The peak at 531.5 eV shows the signal of the metal oxidation state, and the other peak at 532.8 eV belongs to a carbon-oxygen single bond (C-O). The narrow scanning spectrum of the characteristic element Cu 2p is shown in Figure 7c. The binding energy of Cu 2p3/2 peak and Cu 2p1/2 peak is concentrated at 933.8 eV and 953.8 eV, which mainly shows the oxidation state of Cu. The narrow scanning spectrum of characteristic element Zn 2p is shown in Figure 7d, and 1022.4 eV and 1045.1 eV are the characteristic peaks of Zn 2p3/2 and Zn 2p1/2, respectively, which mainly show the oxidation state signal of zinc. XPS results show that stearic acid reacts with the surface of the Cu sample and is successfully modified on the surface of Cu.

### 3.4. Analysis of Surface Corrosion Resistance

Figure 8 is the dynamic potential polarization curves of Cu substrate and superhydrophobic Cu samples treated with different cutter tip distances at 3.5 wt % NaCl solution. Among them, the curve Bare Cu represents the dynamic potential polarization curve of Cu substrate, and the curves MS-25, MS-30, and MS-35 are the dynamic potential polarization curves of superhydrophobic Cu surface with milling tip spacing of 0.25 mm, 0.30 mm, and 0.35 mm respectively. Table 1 shows the detailed data of corrosion potential (*E_corr_*) and corrosion current density (*I_corr_*) measured by Tafel extrapolation method [27], and also gives the corrosion inhibition efficiency (*η*) [28] calculated by Formula (2), where *I*_0 *corr*_ is the corrosion current of Cu substrate and *I_corr_* is the the corrosion current of superhydrophobic Cu sample.
(2)$$\eta =\frac{I_{0corr}-I_{corr}}{I_{0corr}}\times 100\%$$

The results show that the *I_corr_* and *E_corr_* of Cu substrate are 2.92 × 10^−5^ A cm^−2^ and −0.2208 V, respectively. The *I_corr_* and *E_corr_* of superhydrophobic Cu samples (MS-25, MS-30, and MS-35) were 7.29 × 10^−6^ A cm^−2^, 1.47 × 10^−6^ A cm^−2^, 7.51 × 10^−6^ A cm^−2^ and −0.2353 V, −0.2282 V, −0.2258 V, respectively. According to electrochemical theory, the positive values of *I_corr_* and *E_corr_* of the materials with good corrosion resistance are low [29]. The test data show that the *I_corr_* of superhydrophobic Cu samples generally decreases by an order of magnitude compared with that of Cu substrate, which indicates that superhydrophobic Cu samples do have higher corrosion resistance. At the same time, when the cutter tip spacing is 0.30 mm, the superhydrophobic Cu samples have the lowest *I_corr_* and the corrosion inhibition rate reaches 94.97%, and the corrosion protection performance is the best.

In order to characterize the corrosion resistance of superhydrophobic Cu samples, the Nyquist plot, impedance modulus curve, and phase angle plots of each sample in 3.5 wt % NaCl solution are shown in Figure 9a–c. The fitting data are represented by curves, and the scattered points are measured data points. In the Nyquist image (Figure 9a), the diameter of the semicircle curve is related to the charge transfer resistance. The larger semicircle diameter indicates that the sample has good corrosion resistance [30], and the semicircle diameter of the Cu substrate is the smallest, about 300 Ω∙cm^−2^. By contrast, the semicircle diameter of superhydrophobic Cu samples is generally 1–2 orders of magnitude higher, which indicates that superhydrophobic Cu samples have good corrosion resistance. It can also be seen from the Bode image. Generally, the higher the |Z| value in Bode image, the better its corrosion resistance [31]. The data show that the |Z| value of MS-25 and MS-30 are basically the same, and both of them are 1.5 orders of magnitude higher than that of Cu substrate, while the effect of MS-35 is not obvious, only about half an order of magnitude higher. The high phase angle in the high-frequency domain indicates that it has good repulsion performance, and the large modulus in the low-frequency domain indicates that its corrosion resistance is enhanced [32]. Both MS-25 and MS-30 show high phase angles in the low-frequency domain and high-frequency domain. The results show that superhydrophobic Cu samples have good corrosion protection performance, and the corrosion resistance of superhydrophobic Cu samples gradually increases with the decrease in cutter tip distance within a certain range.

Figure 9d,e is the equivalent circuit model (ECs) of Nyquist images of Cu substrate and superhydrophobic Cu samples. In this circuit, the solution resistance is Rs, the film resistance is Rf, and the charge transfer resistance is Rct. Because of the uneven surface of the sample, the constant phase element (CPE) is used instead of the pure capacitor, which is a non-ideal capacitor set for circuit fitting. Compared with the pure capacitor, the constant phase element has a better fitting effect in an equivalent circuit [33]. The Rcts of superhydrophobic Cu samples MS-25, MS-30, and MS-35 are 1898 Ω, 24950 Ω, and 32801 Ω, respectively, which are higher than that of the Cu substrate, which also proves that superhydrophobic Cu samples have good corrosion protection performance. Detailed fitting data are shown in Table 2.

### 3.5. Self-Cleaning Effect

In order to test the surface anti-fouling performance of superhydrophobic Cu samples, the anti-fouling performance of MS-30 superhydrophobic Cu samples was tested, and blue chalk dust was used for the test. The blue chalk is ground into powder and sprinkled on the surface of the superhydrophobic Cu sample, and then the superhydrophobic sample is placed obliquely in a culture dish as shown in Figure 10a. Figure 10b–e shows the process of water droplets rolling off the superhydrophobic Cu surface. It can be seen that, with the rolling of water droplets, the blue chalk dust is absorbed by the water droplets, which leaves the sample surface while rolling. With the increasing number of water droplets, chalk dust can be easily removed as water droplets roll (Figure 10f). The test results showed that the prepared samples had an excellent self-cleaning effect.

### 3.6. Mechanical Durability and Stability

The durability and stability of superhydrophobic surfaces are always important indices to measure the mechanical properties of superhydrophobic surfaces, which greatly affect the practical application of superhydrophobic surfaces. Scratch resistance and friction resistance are usually used to judge the mechanical durability and stability of superhydrophobic surfaces [34,35]. To study the mechanical properties of superhydrophobic Cu samples, a knife scraping test and friction test were selected for MS-30 superhydrophobic Cu samples.

In order to test the surface stability of the superhydrophobic Cu sample, the surface of the sample was scratched with a knife, as shown in Figure 11a. The scratched sample was tilted, and droplets mixed with deionized water and red ink were dripped on the sample surface. Figure 11b–d shows the rolling state of the water droplets. It can be seen that, with the dripping of the water droplets, the rolling of the water droplets is not affected at the intersection of knife marks on the sample surface. Experiments show that the superhydrophobic Cu surface has good scratch resistance and surface stability.

In order to study the mechanical durability of the surface of the superhydrophobic Cu sample, friction test was carried out on the sample. For materials, 2000 # SiC sandpaper, 100 g weight, and a tensimeter were prepared. First, the sandpaper was spread on a horizontal table, connected with the tensimeter, and given a horizontal tension so that the sample could move on the sandpaper at a uniform speed with a distance of pulling of 21 cm per pull. The process was shown in Figure 12a. After being pulled 5 times, 10 times, 15 times, and 20 times, the wear amount of the Cu substrate and superhydrophobic Cu sample were recorded respectively, as shown in Figure 12b.

To judge hydrophobic performance of the superhydrophobic Cu sample after friction and wear, the surface wettability of superhydrophobic Cu sample after the friction test was tested and recorded. Figure 12b shows the CA between the surface of superhydrophobic Cu sample and water under different pulling times. After 5 pulls, the CA of the sample surface was 143.5°, which was 8.9° lower than that of 152.4° before the friction experiment (Figure 4d). After 10, 15, and 20 pulls respectively, the CA of the sample surface is basically stable at around 144°. After 20 pulls, the CA of the sample surface was about 144.5°. It can be observed that the air at the bottom groove of the water drop is not drained. The wetting state of the surface still conforms to the Cassie model, showing a low viscosity state and high CA. With the increase in the number of pulls, the CA of the sample surface shows little change. These results demonstrate that the superhydrophobic Cu surface has good mechanical durability.

When the surface of superhydrophobic samples is abraded by friction, the corrosion resistance of superhydrophobic samples will be seriously affected because the surface structure is damaged. In order to study the corrosion protection performance of the superhydrophobic Cu sample after friction and wear, using 3.5 wt% NaCl solution as electrolyte, the corrosion resistance of MS-30 superhydrophobic Cu sample after 20 pulls in the friction experiment was evaluated (Figure 13). Detailed data are shown in Table 3. The corrosion current density (*I_corr_*) of the superhydrophobic Cu sample after friction testing is 4.70 × 10^−6^ A cm^−2^, which is still one order of magnitude lower than that of Cu substrate (2.92 × 10^−5^ A cm^−2^), and is not much different from that before the friction test (1.47 × 10^−6^ A cm^−2^). The results show that the superhydrophobic Cu surface has excellent mechanical durability, and the worn surface still has good corrosion resistance. This is because when the superhydrophobic Cu surface is rubbed with the SiC sandpaper, the small bumps on the superhydrophobic Cu surface first contact with the sandpaper and rub, and these milled bumps have the strength and hardness of the Cu substrate itself, which are difficult to damage, thus protecting the hydrophobic film layer on the Cu surface from being damaged (Figure 12a). The interaction between the hydrophobic film layer and the microstructure on the Cu surface still forms a good barrier to the sodium chloride aqueous solution, slowing down electric corrosion. Therefore, the superhydrophobic Cu surface can still maintain good corrosion resistance even if it is worn.

## 4. Conclusions

In this paper, a superhydrophobic Cu surface was successfully prepared by milling microstructure on the surface of Cu substrate and modifying with low surface energy substances. The properties of superhydrophobic Cu surface are studied by various tests. The surface microstructure after milling is rectangular protrusions arranged regularly, each protrusion structure has a similar structure and size, and the distance between the protrusions can be controlled by changing the tip distance of milling cutter. After stearic acid modification, the contact angles of the Cu sample surfaces at three milling distances are all greater than 150° in the experiment. Compared with Cu substrate samples, superhydrophobic Cu samples have better corrosion resistance, and the corrosion current density is reduced by one order of magnitude. The surface of superhydrophobic Cu sample also has good self-cleaning, mechanical durability, and stability. After friction and wear, the surface still maintains a high water contact angle and good corrosion resistance. This method is a novel attempt to fabricate superhydrophobic Cu surfaces by mechanical processing, which provides a new means to prepare superhydrophobic Cu with controllable surface morphology. However, this method has certain performance requirements for cutting tools and machine tools. At the same time, the tool wear during processing will affect the size and shape of the surface microstructure of subsequent processing, and how to solve related problems still needs to be further studied.

## Figures and Tables

**Figure 1 materials-15-01939-f001:**
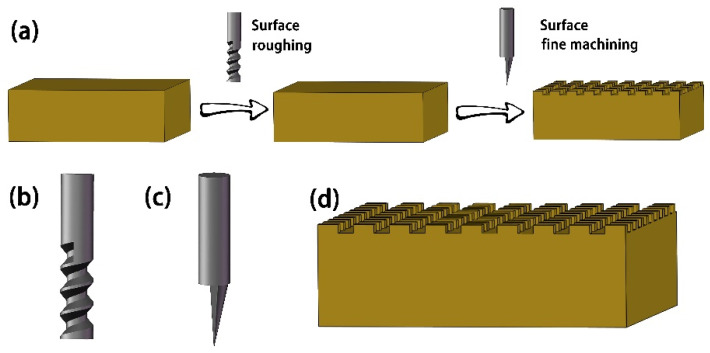
(**a**) Milling process, (**b**) end-milling cutter, and (**c**) flat-bottomed sharp knife. (**d**) Schematic diagram of surface microstructure after milling.

**Figure 2 materials-15-01939-f002:**
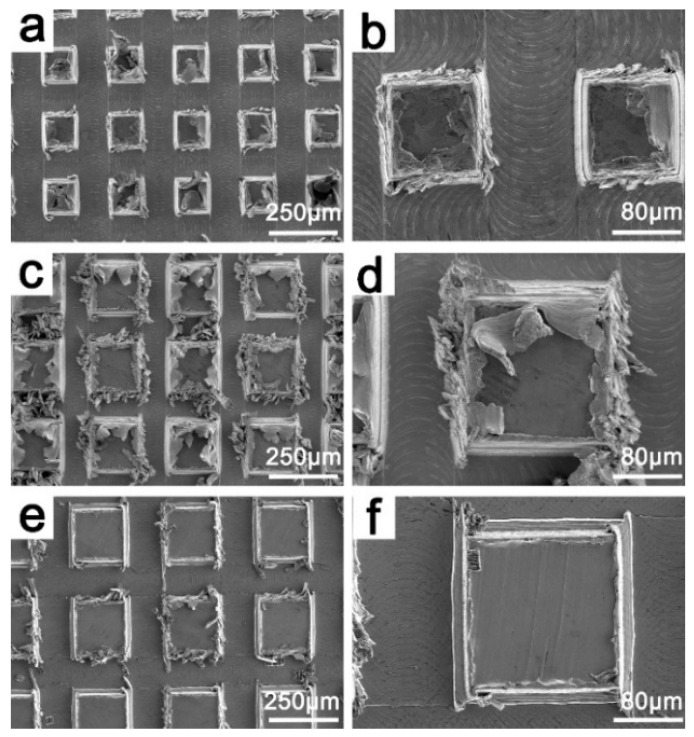
SEM before deburring the surface of milling sample with different cutter tip distances. (**a**,**b**) MS-25, (**c**,**d**) MS-30, (**e**,**f**) MS-35.

**Figure 3 materials-15-01939-f003:**
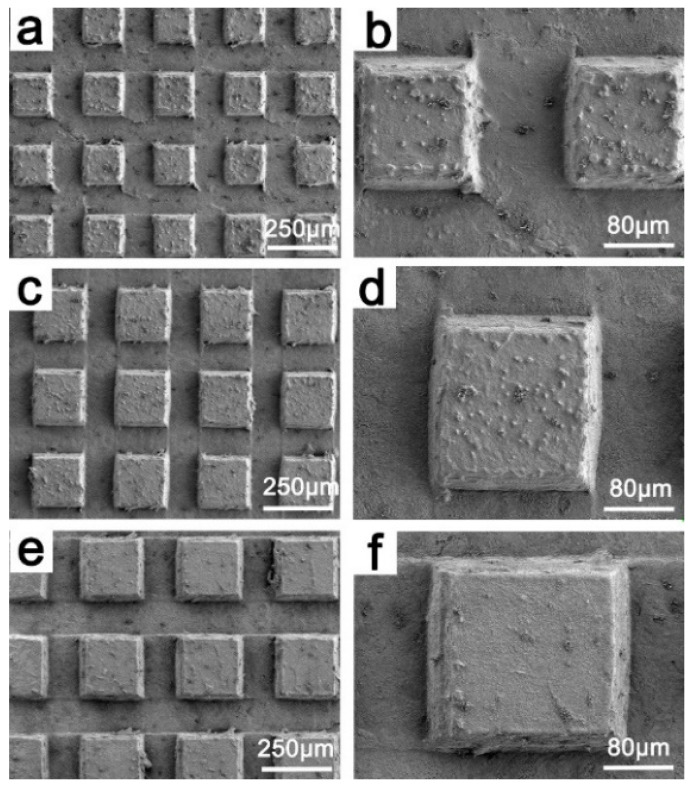
SEM of sample after deburring. (**a**,**b**) MS-25, (**c**,**d**) MS-30, (**e**,**f**) MS-35.

**Figure 4 materials-15-01939-f004:**
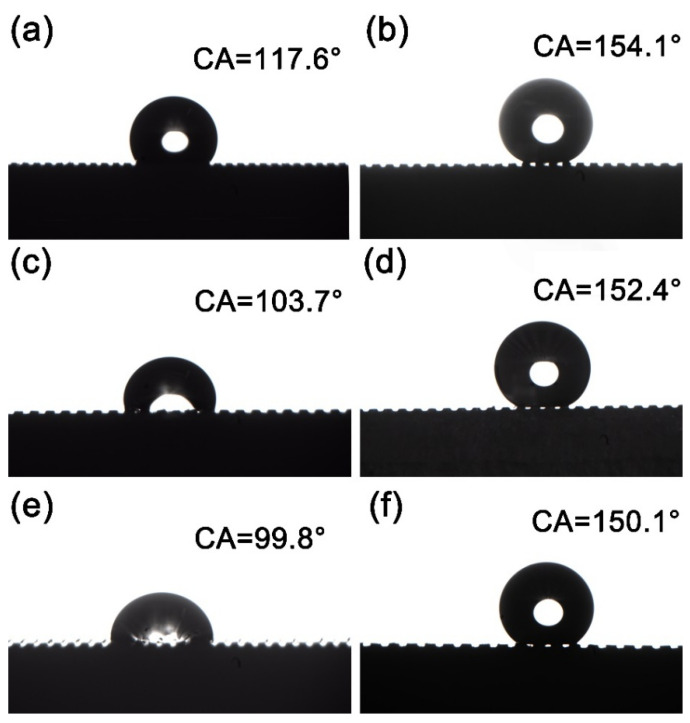
Contact angle images of milling samples with different cutter tip distances. (**a**,**b**) MS-25, (**c**,**d**) MS-30, (**e**,**f**) MS-35.

**Figure 5 materials-15-01939-f005:**
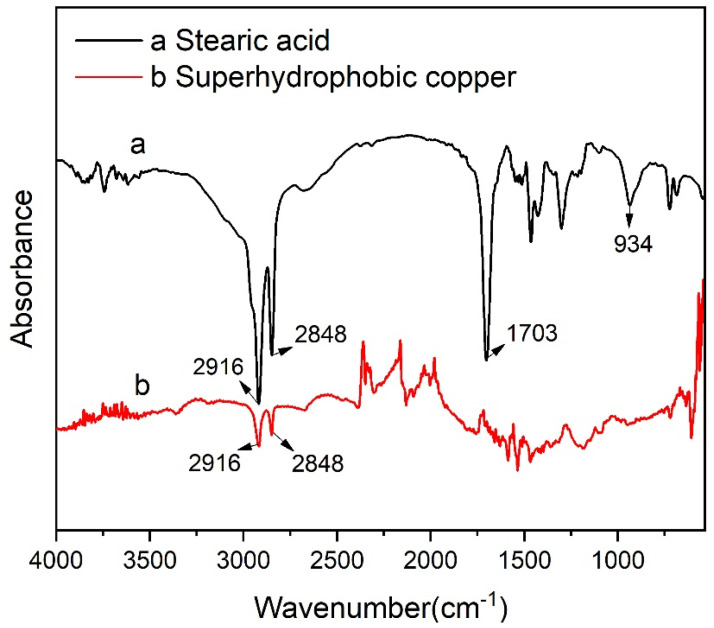
Fourier transform infrared spectra (FTIR) of (**a**) stearic acid and (**b**) superhydrophobic Cu sample surface.

**Figure 6 materials-15-01939-f006:**
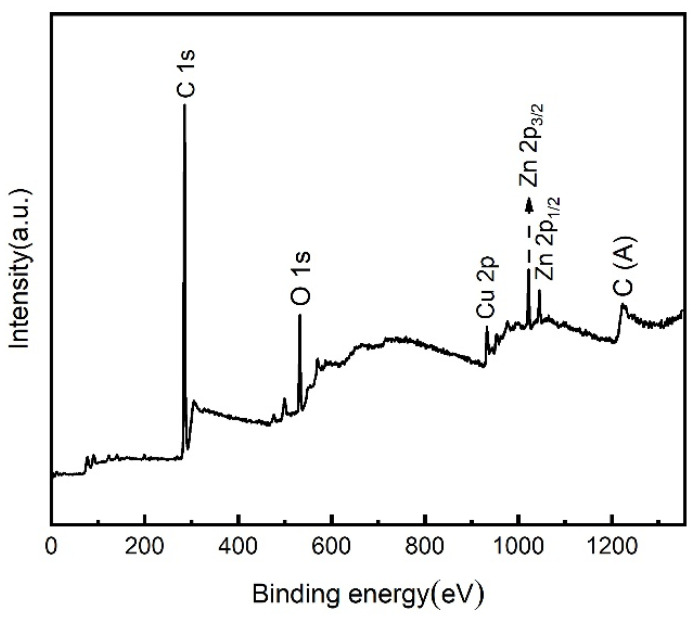
XPS full scanning spectrum of superhydrophobic Cu sample surface.

**Figure 7 materials-15-01939-f007:**
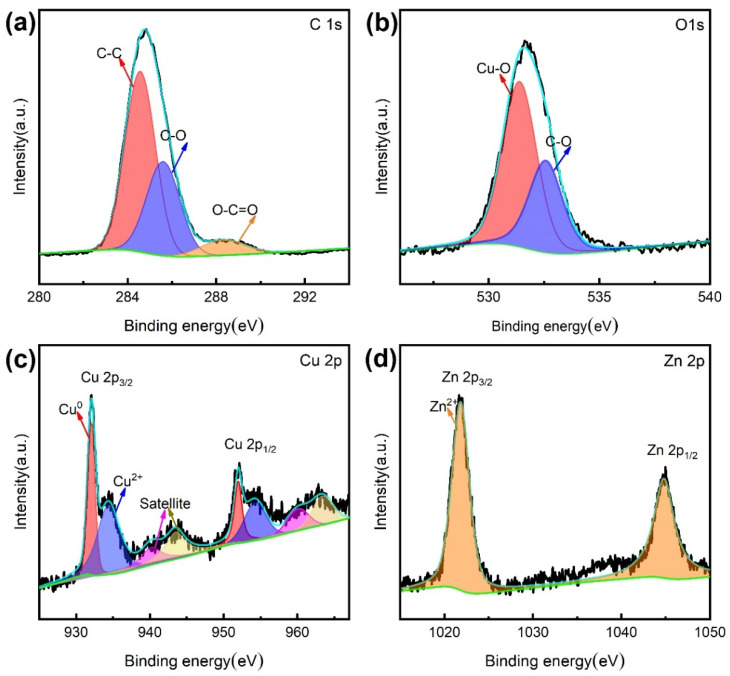
XPS narrow scanning spectrum of superhydrophobic Cu sample. (**a**) C 1s, (**b**) O 1s, (**c**) Cu 2p, (**d**) Zn 2p.

**Figure 8 materials-15-01939-f008:**
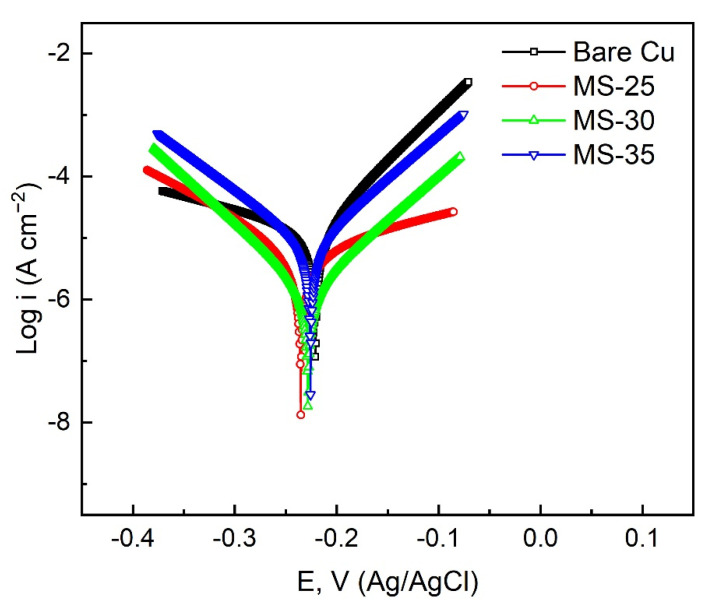
Potentiodynamic polarization curves of different samples in 3.5 wt % NaCl solution.

**Figure 9 materials-15-01939-f009:**
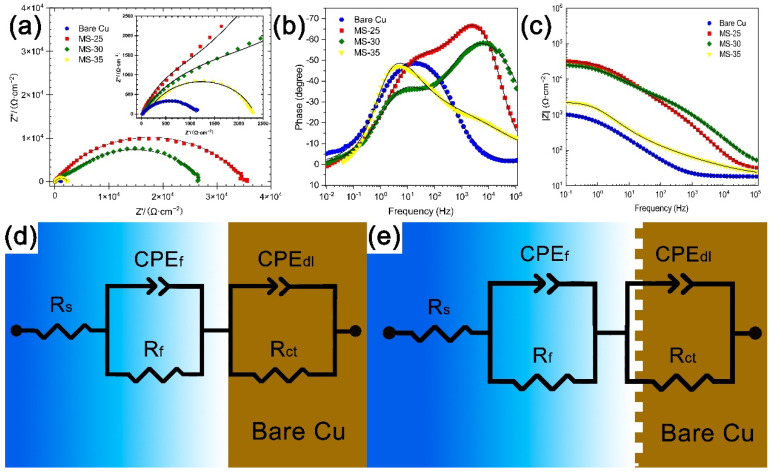
Cu substrate and superhydrophobic Cu samples in 3.5 wt % NaCl solution. (**a**) Nyquist image (**b**), phase angle image (**c**), Bode image, (**d**) equivalent circuit simulation of experimental data of Cu substrate, (**e**) equivalent circuit simulation based on experimental data of superhydrophobic Cu sample.

**Figure 10 materials-15-01939-f010:**
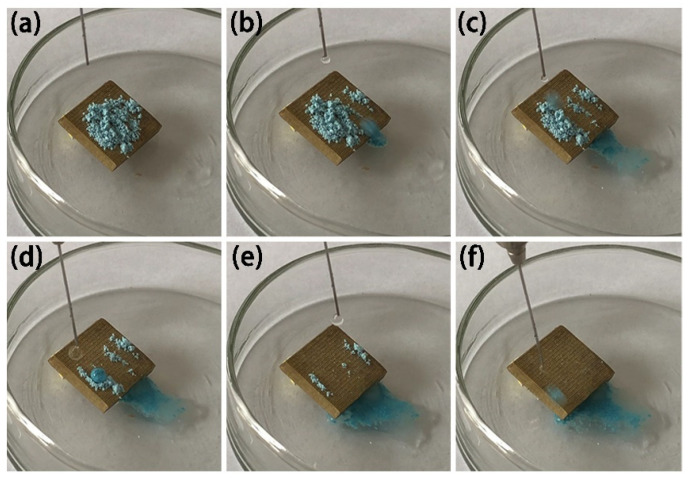
Self-cleaning process image of superhydrophobic Cu sample, powder (**a**) and droplets (**b**–**f**) on sample surface.

**Figure 11 materials-15-01939-f011:**
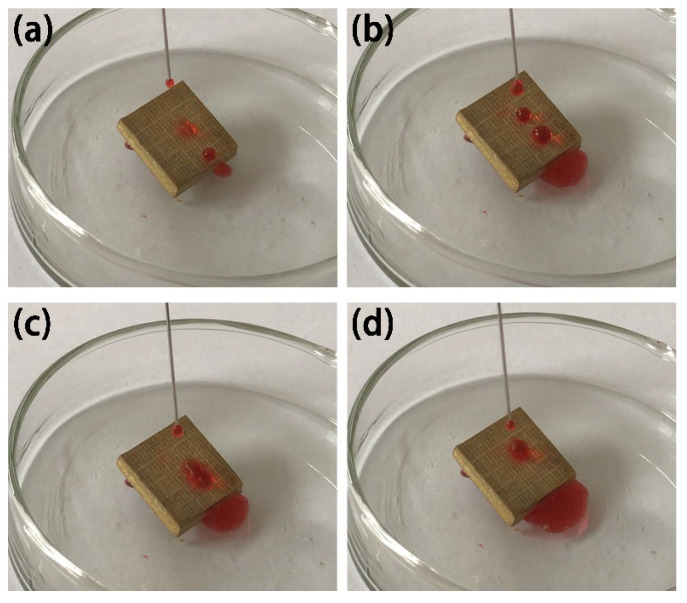
Knife scraping test image of a superhydrophobic Cu sample. (**a**–**d**) droplets on sample surface.

**Figure 12 materials-15-01939-f012:**
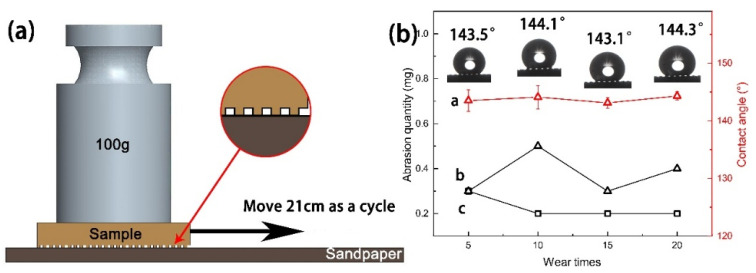
(**a**) Schematic diagram of friction test, (**b**) variation diagram of wear amount and contact angle (CA): a is the surface CA change curve of superhydrophobic Cu sample; b is the surface wear curve of superhydrophobic Cu sample; c is the surface wear curve of Cu substrate.

**Figure 13 materials-15-01939-f013:**
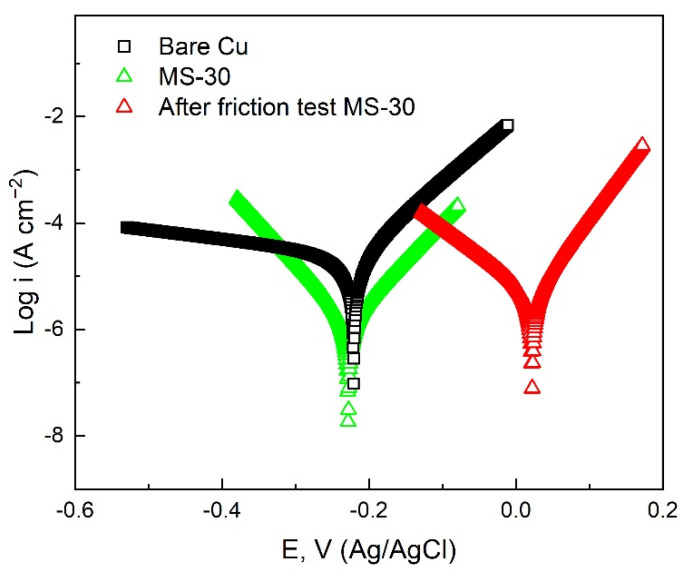
Potentiodynamic polarization curves of Cu substrate, superhydrophobic Cu sample before wear, and superhydrophobic Cu sample after wear in 3.5 wt % NaCl solution.

**Table 1 materials-15-01939-t001:** Electrochemical data of the polarization curves obtained from the samples.

Sample	*E_corr_* (V vs. Ag/AgCl)	*I_corr_* (A cm^−2^) (A/cm^2^)	*η* (%)
Bare Cu	−0.2208	2.92 × 10^−5^	––––
MS-25	−0.2353	7.29 × 10^−6^	75.03
MS-30	−0.2282	1.47 × 10^−6^	94.97
MS-35	−0.2258	7.51 × 10^−6^	74.28

**Table 2 materials-15-01939-t002:** Equivalent circuit fitting results of the EIS data.

Sample	Rs(Ω cm^2^)	CPE_f_ (µF/cm^−2^∙s^(α−1)^)	Rf(Ω cm^2^)	CPE_dl_ (µF/cm^−2^∙s^(α−1)^)	Rct(Ω cm^2^)
Bare Cu	18.02	18.00	773.5	10.86	1136
MS-25	25.14	48.44	852.3	2.97	32,801
MS-30	24.84	13.92	2248	5.44	24,950
MS-35	15.92	164.83	457.8	36.10	1898

**Table 3 materials-15-01939-t003:** Electrochemical data of the polarization curves obtained from the samples.

Sample	*E_corr_* (V vs. Ag/AgCl)	*I_corr_* (A cm^−2^)	*η* (%)
Bare Cu	−0.2208	2.92 × 10^−5^	––––
MS-30	−0.2282	1.47 × 10^−6^	94.97
After friction test MS-30	−0.2188	4.70 × 10^−6^	83.90

## Data Availability

The data presented in this study are available on request from the corresponding author.

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
