# Peer review of "Study on Preparation of Superhydrophobic Copper Surface by Milling and Its Protective Performance"

_materials, 2022, doi:10.3390/ma15051939_

Round 1

Reviewer 1 Report

The authors created microstructures on Cu alloys using different depths and modified the surface using stearic acid. I believe this work has enough novelty to be published in the Materials journal. Nevertheless, the following issues must be considered before considering this paper for publications.

1- The authors must always compare with the bare Cu surface with and without stearic acid.

2- The authors need to show the effect of time of the WCA.

3- The authors need to show the effect of Time on the EIS and Tafel analysis.

4- The EIS Nyquist plots must have orthogonal axes.

5- Please remove the Warburg diffusion element from the equivalent circuits as it is not show in the plot, so we can not bring it to the equivalent cirecuit.

6- The authors need to show the effect of pH on the the stability of the superhydrophobicity.

7- The authors need to show the effect of UV on the stability of the supherhydrophobicity.

8- The reference electrode is not mentioned on figure 8. Also, Ecorr and Icorr must be italic and units must follow IUPAC i.e. A cm-2 not A/cm2.

9- Current axis in figure 8 should be Log i (A cm-2) and potential should be E, V (Ag/AgCl). Also, the same should be done for figure 13.

10- The EIS measrurements should be for systems at steady state. However, the Cu alloy in NaCl will suffer from pitting which is a stichastic event. How long the experiment takes to measure the EIS plot compared to the inititation of the first pit.

11- The  polarization curves should be extended to measure the pitting potential for the four different electrodes.

12- The authors need to calculate the percentage of air trapped under the water droplet based on the water contact angle.

13- The authors need to do SEM/EDX after the corrosion test to see whether there is enriching on one element on the surface or not.

14- The authors also need to show the effect of Temperature on the WCA.

15- The authors also must show the contact angle hysteresis and sliding behavior of water droplets.

16- The following reference should be added to the introduction

Corrosion behavior of superhydrophobic surfaces: A review AMA Mohamed, AM Abdullah, NA Younan Arabian journal of chemistry 8 (6), 749-765

Author Response

  • The authors must always compare with the bare Cu surface with and without stearic acid.

Response: Thank you for your valuable advice. Many related research reports present that compared with bare copper, Superhydrophobic surfaces can greatly enhance corrosion protection for the copper matrix after modifying with low surface energy materials like stearic acid. Our previous research aims to prove the feasibility of the milling and so we neglect the comparison of the water contact angle of bare Cu surface with and without stearic acid. Due to the urgency of time and closing of our laboratory for the epidemic, we haven’t completed the comparison of these two samples. If the reviewer insists on the comparison, we will submit our research finding later.

2- The authors need to show the effect of time of the WCA.

Response: Thank you for your valuable advice. When we finished our experiment, the prepared sample has been placed in the lab for several months, it still maintained its superhydrophobicity. But due to our due to our negligence, we didn’t record the effect of time of WCA. Now because of the urgency of time and closing of our laboratory for the epidemic, we couldn’t resume the experiment in a short time and supplement your suggestions. We are very sorry for this.

3- The authors need to show the effect of Time on the EIS and Tafel analysis.

Response: Thank you for your valuable advice. Also for the urgency of time and closing of our laboratory for the epidemic, we couldn’t resume the experiment in a short time and supplement your suggestions. We are very sorry for this.

4- The EIS Nyquist plots must have orthogonal axes.

Response: Thank you for your valuable advice. The picture has been revised.

5- Please remove the Warburg diffusion element from the equivalent circuits as it is not show in the plot, so we can not bring it to the equivalent cirecuit.

Response: Thank you for your valuable comments. Now the Warburg diffusion element has been removed from the equivalent circuit.

6- The authors need to show the effect of pH on the the stability of the superhydrophobicity.

Response: Thank you for your valuable advice. Also for the urgency of time and closing of our laboratory for the epidemic, we couldn’t resume the experiment in a short time and supplement your suggestions. We are very sorry for this.

7- The authors need to show the effect of UV on the stability of the supherhydrophobicity.

Response: Thank you for your valuable advice. Also for the urgency of time and closing of our laboratory for the epidemic, we couldn’t resume the experiment in a short time and supplement your suggestions. We are very sorry for this.

8- The reference electrode is not mentioned on figure 8. Also, Ecorr and Icorr must be italic and units must follow IUPAC i.e. A cm-2 not A/cm2.

Response: Thank you for your valuable comments. This is my mistake. Now, the names of the horizontal and vertical axes in Figure 8 and Figure 13 have been modified and supplemented. At the same time, the Ecorr and Icorr in the article have also been changed to italics.

9- Current axis in figure 8 should be Log i (A cm-2) and potential should be E, V (Ag/AgCl). Also, the same should be done for figure 13.

Response: Thank you for your valuable comments. Now, the names of the horizontal and vertical axes in Figure 8 and Figure 13 have been modified and supplemented.

10- The EIS measrurements should be for systems at steady state. However, the Cu alloy in NaCl will suffer from pitting which is a stichastic event. How long the experiment takes to measure the EIS plot compared to the inititation of the first pit.

Response: The experiment takes a total of 33 minutes. As shown in Figure 1 above, the intersection of horizontal and vertical dash-dot lines is the specific critical potential Eb for pitting, in which Eb of Cu substrate and MS-35 samples occurs around 620s, and Eb of MS-25 and MS-30 samples occurs around 660s. Therefore, compared with the initial position of the first pit, the experiment takes 22min to 23min to measure the fitting circuit diagram.

11- The  polarization curves should be extended to measure the pitting potential for the four different electrodes.

Response: Thank you for your valuable advice. As shown in Figure 2 above, the left side of Figure 2 is a schematic diagram of a three-electrode system, and the right side of Figure 2 is a schematic diagram of a four-electrode system. Three-electrode system is the most commonly used method in electrochemical testing. Compared with the three-electrode system, the four-electrode system separates WS from W. Although it can make more accurate measurement, this system is relatively less used in electrochemical measurement, and it is mainly used for accurate measurement of solution resistance or metal surface resistance (solid-state battery). In addition, the measurement of the four-electrode system needs electrochemical electrolytic cell devices with multiple interfaces, and the previous experiment did not involve the requirement of the four-electrode system experiment, resulting in the electrolytic cell devices in the laboratory only supporting three-system electrodes. Considering the above two factors, we suggest that we continue to use the three-electrode system for measurement in this article. We will update the experimental equipment for further research, and hope you can adopt it.

12- The authors need to calculate the percentage of air trapped under the water droplet based on the water contact angle.

Response: Thank you for your valuable suggestions. We have revised and supplemented the article.

13- The authors need to do SEM/EDX after the corrosion test to see whether there is enriching on one element on the surface or not.

Response:Thank you for your valuable advice. Also for the urgency of time and closing of our laboratory for the epidemic, we couldn’t resume the experiment in a short time and supplement your suggestions. We are very sorry for this.

14- The authors also need to show the effect of Temperature on the WCA.

Response: Thank you for your valuable advice. Also for the urgency of time and closing of our laboratory for the epidemic, we couldn’t resume the experiment in a short time and supplement your suggestions. We are very sorry for this.

15- The authors also must show the contact angle hysteresis and sliding behavior of water droplets.

Response: Thank you for your valuable advice. Also for the urgency of time and closing of our laboratory for the epidemic, we couldn’t resume the experiment in a short time and supplement your suggestions. We are very sorry for this.

16- The following reference should be added to the introduction

Corrosion behavior of superhydrophobic surfaces: A review AMA Mohamed, AM Abdullah, NA Younan Arabian journal of chemistry 8 (6), 749-765

Response: Thank you for your valuable comments. We have read this article and added it to the references in the introduction.

Reviewer 2 Report

Interesting research. Easy to read. Suggestions for improving the article are the following:

Comment 1

I think that it is better to state the surname of the first author when directly citing previous research.

For example, in the text, instead of:

H.M. et al. took advantage of the dislocation defects in crystalline metals which are more likely to be eroded, and prepared the superhydrophobic Cu surface with micro-nano structure by dislocation etching, shot peening and stearic acid treatment, and the highest contact angle reached 153°. Because of the uncontrollable chemical etching surface, shot peening is used to densify the surface after etching, making a cumbersome process to roughen the surface [17].

to be:

Forooshani et al. [17] took advantage of the dislocation defects in crystalline metals which are more likely to be eroded, and prepared the superhydrophobic Cu surface with micro-nano structure by dislocation etching, shot peening and stearic acid treatment, and the highest contact angle reached 153°. Because of the uncontrollable chemical etching surface, shot peening is used to densify the surface after etching, making a cumbersome process to roughen the surface.

Comment 2

The literature (References section in the article) is not listed in the required format. Correct the way the literature is cited (Abbreviated Journal Name, etc.). https://www.mdpi.com/journal/materials/instructions

Comment 3

In the last paragraph of the Introduction section, before stating the goal of the paper, you must write the shortcomings of previous research. In a few sentences, point out the shortcomings of previous research. Write the goal of your research and present the scientific hypotheses. Additionally highlight the scientific contribution of your research.

Comment 4

For the materials used in the research, show the mechanical, physical and chemical characteristics (Materials subsection).

Comment 5

The authors state the following: Firstly, the surface of Cu block is roughly machined with an end-milling of 4 mm diameter (Fig. 1b), so that the surface of the sample is parallel and flat. In my opinion, it is very dangerous to state that something is parallel and flat. These are geometric product specifications (GPS). Show the values of parallelism and flatness in the corrected article.

Comment 6

How did you select the milling parameters? Why these values? Elaborate further in the article.

Comment 7

A great lack of research that has not considered the errors of measurements and their impact on the results obtained. Assess the measurement uncertainty of your results. Further discuss the obtained values of the measured uncertainty. What’s more, sensitivity analysis and uncertainty analysis would be good to do.

Comment 8

Further analyze and discuss the possibilities of practical application of your methodology.

Comment 9

In the section Conclusions, it should be emphasized: the scientific contribution of research; limitations of applied methodology and future research.

Author Response

Comment 1

I think that it is better to state the surname of the first author when directly citing previous research.

For example, in the text, instead of:

H.M. et al. took advantage of the dislocation defects in crystalline metals which are more likely to be eroded, and prepared the superhydrophobic Cu surface with micro-nano structure by dislocation etching, shot peening and stearic acid treatment, and the highest contact angle reached 153°. Because of the uncontrollable chemical etching surface, shot peening is used to densify the surface after etching, making a cumbersome process to roughen the surface [17].

to be:

Forooshani et al. [17] took advantage of the dislocation defects in crystalline metals which are more likely to be eroded, and prepared the superhydrophobic Cu surface with micro-nano structure by dislocation etching, shot peening and stearic acid treatment, and the highest contact angle reached 153°. Because of the uncontrollable chemical etching surface, shot peening is used to densify the surface after etching, making a cumbersome process to roughen the surface.

Response: Thank you for your valuable advice. It was my negligence and has been revised.

Comment 2

The literature (References section in the article) is not listed in the required format. Correct the way the literature is cited (Abbreviated Journal Name, etc.). https://www.mdpi.com/journal/materials/instructions

Response: Thank you for your valuable advice, which has been revised as required.

Comment 3

In the last paragraph of the Introduction section, before stating the goal of the paper, you must write the shortcomings of previous research. In a few sentences, point out the shortcomings of previous research. Write the goal of your research and present the scientific hypotheses. Additionally highlight the scientific contribution of your research.

Response: Thank you for your valuable advice. The introduction has been modified and supplemented.

Comment 4

For the materials used in the research, show the mechanical, physical and chemical characteristics (Materials subsection).

Response: Thank you for your valuable advice, which has been supplemented by the material subsection section.

Comment 5

The authors state the following: Firstly, the surface of Cu block is roughly machined with an end-milling of 4 mm diameter (Fig. 1b), so that the surface of the sample is parallel and flat. In my opinion, it is very dangerous to state that something is parallel and flat. These are geometric product specifications (GPS). Show the values of parallelism and flatness in the corrected article.

Response: Thank you for your valuable advice. The expression here in the article is biased. In the experiment, because of the high machining accuracy, the purpose of rough machining is to make the surface of the sample and the milling cutter's machining plane in the same horizontal position, so as to avoid the inconsistency of the surface groove depth caused by the flat-bottomed sharp knife in finishing. This part has been modified in the article.

Comment 6

How did you select the milling parameters? Why these values? Elaborate further in the article.

Response: The experimental parameters refer to the related research of Rahman, M.A et al. [23] and Pratap T et al. [24], and are adjusted according to the actual situation.

Comment 7

A great lack of research that has not considered the errors of measurements and their impact on the results obtained. Assess the measurement uncertainty of your results. Further discuss the obtained values of the measured uncertainty. What’s more, sensitivity analysis and uncertainty analysis would be good to do.

Response: Thank you for your valuable advice. The influence of the size, shape and other factors of the surface microstructure of the sample after milling on the subsequent superhydrophobic contact angle and rolling angle will be further studied in the follow-up articles. In this paper, when measuring the static contact angle, in order to ensure the accuracy of the test, 6μl of static water droplets were placed in five different positions of each sample for contact angle test, and the average value of five groups of contact angle values was taken. The influence of uncertain factors on the experimental results has been eliminated as much as possible.

Comment 8

Further analyze and discuss the possibilities of practical application of your methodology.

Response: Thank you for your valuable advice. We have revised and supplemented and modified in the last paragraph of the introduction.

Comment 9

In the section Conclusions, it should be emphasized: the scientific contribution of research; limitations of applied methodology and future research.

Response: Thank you for your valuable advice. The conclusion has been modified and supplemented.

Round 2

Reviewer 2 Report

Based on a detailed review of the corrected manuscript, I conclude that the authors have made updates and corrections. I suggest accepting the article in its current form.